# The Coding Mitogenome of the Freshwater Crayfish *Pontastacus leptodactylus* (Decapoda:Astacidea:Astacidae) from Lake Vegoritida, Greece and Its Taxonomic Classification

**DOI:** 10.3390/genes14020494

**Published:** 2023-02-15

**Authors:** Maria V. Alvanou, Apostolos P. Apostolidis, Athanasios Lattos, Basile Michaelidis, Ioannis A. Giantsis

**Affiliations:** 1Department of Animal Science, Faculty of Agricultural Sciences, University of Western Macedonia, 53100 Florina, Greece; 2Laboratory of Ichthyology & Fisheries, Department of Animal Production, Faculty of Agriculture, Forestry and Natural Environment, Aristotle University of Thessaloniki, 54124 Thessaloniki, Greece; 3Laboratory of Animal Physiology, Department of Zoology, Faculty of Science, School of Biology, Aristotle University of Thessaloniki, 54124 Thessaloniki, Greece

**Keywords:** *Pontastacus leptodactylus*, phylogeny, mitochondrial genome, Vegoritida, freshwater crayfish

## Abstract

*Pontastacus leptodactylus* (Eschscholtz, 1823) (Decapoda:Astacidea:Astacidae) constitutes an ecologically and economically highly important species. In the present study, the mitochondrial genome of the freshwater crayfish *P. leptodactylus* from Greece is analyzed for the first time, using 15 newly designed primer pairs based on available sequences of closely related species. The analyzed coding part of the mitochondrial genome of *P. leptodactylus* consists of 15,050 base pairs including 13 protein-coding genes (PCGs), 2 ribosomal RNA gene (rRNAs), and 22 transfer RNA genes (tRNAs). These newly designed primers may be particularly useful in future studies for analyzing different mitochondrial DNA segments. Based on the entire mitochondrial genome sequence, compared to other haplotypes from related species belonging in the same family (Astacidae) available in the GenBank database, a phylogenetic tree was constructed depicting the phylogenetic relationships of *P. leptodactylus*. Based on the results, the genetic distance between *Astacus astacus* and *P. leptodactylus* is smaller than the genetic distance between *Austropotamobius pallipes* and *Austropotamobius torrentium*, despite the fact that the latter two are classified within the same genus, questioning the phylogenetic position of *A. astacus* as a different genus than *P. leptodactylus*. In addition, the sample from Greece seems genetically distant compared with a conspecific haplotype available in the GenBank database, possibly implying a genetic distinction of *P. leptodactylus* from Greece.

## 1. Introduction

The narrow-clawed crayfish *Pontastacus leptodactylus* (Eschscholtz, 1823) (Decapoda:Astacidea:Astacidae), apart from representing a keystone species operating as a benthic scavenger, is, in parallel, a food product of high economic value. Although it is characterized as least concern in IUCN [1], there are several studies addressing its population decline within specific basins [2,3,4]. Furthermore, in Lake Vegoritida, as well as in Lake Polifitou in northern Greece, such phenomena have been observed leading to the periodical prohibition of crayfishing [5]. The narrow-clawed crayfish can, generally, reach a relatively large size and can tolerate altered water conditions [5,6], as it has been found in brackish waters as well, implying its tolerance towards fluctuating salinity levels [7]. Additionally, it seems to be more tolerant towards diseases in comparison with other indigenous species [8].

Four freshwater crayfish species are found in Greece. More specifically, there are three indigenous (*Astacus astacus*, *P. leptodactylus*, *Austropotamobius torrentium*) and one alien (*Pacifastacus leniusculus*) species. Previous studies concerning its geographic spread report that *P. leptodactylus* is native to the Evros region in Greece [9,10]. However, due to its high economic significance and export orientation, many translocations and introductions of *P. leptodactylus* have been carried out in new habitats where it had not been recorded in the past. Lakes Vegoritida and Volvi in north Greece are included in the above category, as *P. leptodactylus* was recently recorded there, probably reflecting recent translocation events [5]. It should be also highlighted that this species presents a high aquaculture potential in north Greece [5]. An interesting question rises regarding translocation events that requires the investigation of the genetic influence (if existing) on the populations due to the different environmental conditions existing in the new habitats or the co-occurrence with other species, and, subsequently, the hybrids that may exist. Hybrids have been observed in a previous study, indicating the high possibility they also exist in several ecosystems that freshwater crayfish inhabit [11].

*P. leptodactylus* is the second most widespread freshwater crayfish indigenous species in Europe, widely distributed and found in 32 countries [12]. More specifically, it is indigenous to Europe and Anatolia [13] and its native range is associated with the Ponto–Caspian water bodies [14]. Despite its economic value, its broad distribution, and its declining populations, the phylogenetic relationships between populations of this species with other members of the family Astacidae are not completely clear, resulting in confusion regarding its systematic classification. The above situation is reflected in the different proposed names such as *P. leptodactylus*, *Astacus leptodactylus*, or as *Astacus leptodactylus*-complex species [6,15,16,17,18,19]. Additionally, many different assumptions for the number of subgenera and subspecies have been proposed in an attempt to define the taxonomic position of the group [5,16,17,20,21]. Its most prevalent classification at the genus level is considered the *P. leptodactylus* [19]. However, it is worth mentioning that the majority of above interferences are based on specific mitochondrial genes’ analyses and a more comprehensive analysis, including the complete mitochondrial genome, is needed to enlighten the taxonomy.

Mitochondrial genomes have been widely used for phylogenetic analyses, being considered as a very useful tool for reconstructing phylogenies. In animals, the typical mitochondrial genome is a circular duplex molecule, which contains 13 protein-coding genes (PCGs), 22 transfer RNA (tRNA) genes, 2 ribosomal RNA (rRNA) genes, and the control region (D-loop), which is usually an AT-rich noncoding sequence [22]. Previous genetic studies on *P. leptodactylus* were mainly focused on genetic markers such as allozymes, single mitochondrial genes, and microsatellites [11,23,24,25], with a remarkable absence of complete mitochondrial-genome-based studies. Previous phylogenetic studies based on COI revealed two different phylogroups, one including European populations and another including Asian populations [23]. Furthermore, among the species, two subspecies have been identified in Turkey, with one distributed in the western and central Anatolia region of Turkey, and the second distributed in the eastern Thrace and Marmara regions of Turkey [12]. The lack of a comprehensive phylogenetic study of *P. leptodactylus* is also highlighted from more recent studies [26]. However, in Greece, scarce data exist regarding the genetic identity of this species.

DNA barcoding, especially when cytochrome oxidase 1 (cox1) is used as a reference [27], has raised some concerns [28]. More specifically, it has been revealed that for some taxonomic groups, including crustaceans [29], DNA barcoding is not a completely reliable method. The above is mainly supported by the uncertainty existing towards the impact of mitochondrial DNA copies in the nuclear genome (NUMTS). NUMTs are referred to as copies of mitochondrial genomic elements that are transferred into the nuclear genome [30]. NUMTS have also been identified in freshwater crayfish including in the *Cherax* genus [27,31]. The combination of limited genomic information with the limited data regarding the molecular basis of gene expression and phenotypic variation may pose obstacles towards the aquaculture perspective of the species [32]. One major issue in phylogenetic analysis is data exclusion. Most of the studies are based on one or two mt genes or exclude some protein-coding genes from the whole mt sequence [33]. Therefore, whole mitogenome analysis regarding crustacean species may provide remarkable results towards understanding their phylogenetic gaps. In the past, whole mt genome phylogenies resolved problems existing in the phylogenetic groups of all arthropods such as intraordinal relationships [34].

Hence, taken all together, it is clear that a more robust analysis, including more mtDNA concatenated sequences or even the whole mitogenome, will provide a clearer picture regarding the phylogenetic position of the narrow-clawed crayfish inhabiting Greece. Therefore, the aim of the present study was firstly to analyze the entire mitochondrial genome of the specific species collected from Greece for the first time, in an attempt to further clarify its systematic classification. The development of a low cost and easy-to-use tool for the whole mitogenome analysis will be the first step towards future studies regarding the reconstruction of the phylogeny of the family Astacidae. More specifically, the phylogenetic investigation of *P. leptodactylus* species complex will provide information to fill the existing gap and reveal more details regarding the species status after and before translocations, as well as shedding light on the investigation of existence of specific species and subspecies within this genus.

## 2. Materials and Methods

Genomic DNA was extracted from the abdominal muscle tissue sample of *P. leptodactylus* from 5 crayfish individuals caught in Lake Vegoritida, northwestern Greece. The samples were immediately embedded in liquid nitrogen prior to DNA extraction. Forty (40) μg muscle tissue were obtained for DNA isolation, performed using the kit Nucleospin tissue (Machenery Nagel, Duren, Germany) following the recommendations of the manufacturer. The concentration and the purity of the DNA were evaluated using a Q5000 microvolume spectrophotometer (Quawell Technology Inc, San Jose, CA, USA). Fifteen primer sets were designed using the Primer 3 software (Primer3_masker, Tartu, Estonia) [35] according to mtDNA sequences obtained from NCBI of members of Astacidae family (Table 1). More specifically, the sequences with GenBank accession numbers KX279349.1 and KX279347.1 were utilized. The above accession numbers correspond to *A. leptodactylus* and *A. astacus*, respectively. 

The 15 mtDNA segments were amplified in 15 separate PCRs for each individual. Each 20 μL PCR reaction contained 10 μL FastGene Taq 2X Ready Mix (NIPPON Genetics, Duren, Germany), 0.6 μL of each primer (10 μM), 1 μL of DNA (50 ng/μL), and ultrapure water up to the final volume. The conditions of each reaction were 95 °C for 3 min, 94 °C for 30 s, annealing temperature (Table 1) for 40 s, 72 °C for 50 s, and a final amplification step at 72 °C for 5 min. The PCR products were run on a 2% agarose for 40 min at 100 V. The fifteen PCR products from one individual are depicted in Figure 1. The DNA ladder used was 100 bp (H3 RTU, GeneDireX, Taoyuan, Taiwan). After purification of the PCR products using the commercial NucleoSpin Gel and PCR clean up kit (MAcherey-Nagel, Duren, Germany), the purified products were bidirectionally sequenced applying the Sanger methodology. 

The nucleotide composition and the relative synonymous codon usage (RSCU) were determined using MEGAΧ [36]. AT skew = (A − T)/(A + T) and GC skew = (G − C)/(G + C) analysis were performed in order to describe the base composition of the coding mitogenome.

Finally, using the MEGAX software [36], the results from sequencing were aligned using the ClustalW algorithm and a maximum likelihood (ML) phylogenetic tree was created including five sequences retrieved from GenBank with accession numbers: NC_033509.1, NC_033504.1, NC_026560.1, KX279349.1, and KX279347.1. Only sequences of the same lengths were included in the analysis. For ML tree construction with MEGAX, the best-fit substitution model (GTR  +  G) was determined applying the Akaike information criterion (AIC). The tree was visualized with MEGAX. The annotation of the coding mitogenome of *P. leptodactylus* was conducted according to the sequence with accession number KX279349, which is deposited in the GenBank database. The preliminary annotation was conducted using MITOS, which revealed the locations of the protein-coding and rRNA genes. The start and stop codons were identified using ORF finder and Blastn of NCBI [37].

## 3. Results

Overall, 15 mtDNA PCR products from *P. leptodactylus* sample were amplified (Figure 1). However, due to sequencing problems attributed to lack of reference sequences of D-loop, and, thus, uncertainties in annotation, 14 of them were successfully sequenced. The characterized haplotype was deposited in the GenBank database and assigned the accession number OQ131122.

### 3.1. Genome Composition

Hence, we present here the complete mitochondrial coding of *P. leptodactylus* genome. The coding part of the mitochondrial genome of *P. leptodactylus*, according to the sequencing results, consists of 15,050 base pairs. The part of the control region missing is evaluated to be approximately 1500–2000 base pairs, according to the amplified PCR product, which is in line with the mitochondrial sequences existing in the NCBI. Within the mitochondrial genome, 13 protein-coding genes, 2 ribosomal RNA, and 22 transfer RNAs are included (Table 2). Eleven out of thirteen PCGs are coded on the plus (+) strand while the other two are on the minus (−) strand. Both rRNAs and 15 tRNAs are coded on the + strand, while 7 tRNAs are on the minus. Furthermore, all the genes are closely aligned with only a small number of overlapping base pairs. The schematic illustration of the mitochondrial genome of *P. leptodactylus* is depicted in Figure 2. Thus, from the complete mitochondrial genome, we are probably missing the D-loop according to the typical mitochondrial genome structure observed in animals. The D-loop in length is approx. 1500–2000 bp (as depicted in Figure 1, in number 5 well), which is considered as a moderate or slightly big size for this region. However, fluctuations regarding the size of the D-loop are common among species as, generally, this region is considered to exhibit the most significant length variation in the mitogenome [38].

The base composition of the coding mitogenome of *P. leptodactylus* presents a strong A + T bias (71.4%). More specifically, the base composition is found to be T: 39.1%, C: 11.3%, A: 32.3%, and G: 17.3%. Additionally, a negative AT skew (−0.095) and a positive GC skew (0.210) are observed.

### 3.2. Protein-Coding Genes, tRNAs, and rRNAs

The PCG region is calculated to be 11,153 bp in length, which corresponds to 74.11% of the whole coding mitogenome of *P. leptodactylus*. For the tRNA and rRNA region, the length is found to be 1427 bp and 2139 bp, corresponding to 9.48% and 14.57%, respectively, of the whole coding mitogenome of narrow-clawed crayfish. The length of the 13 PCGs ranges between 159 bp (*atp8*) and 1734 bp (*nad5*), while the length of 22 tRNAs ranges between 62 bp and 70 bp. Eleven of the PCGs are initiated by a canonical ATN codon while the other two (*atp8*; *nad2*) are initiated by near-cognate GTG codon. Furthermore, ten of them terminate with a typical TAA stop codon while the other three (*cox1*; *atp8*; *cytb*) terminate with TTA, TAG, and ATT, respectively. The rRNA genes (*rrnS*; *rrnL)* are located between *trnN* and *trnV* and between *trnV* and *trnL1*, and they are 814 bp and 1379 bp in length, respectively. The D-loop region is located between *trnE* and *trnQ* and it is estimated to be between 1500 and 2000 bp in length.

The average codon frequencies of the PCGs were calculated. From the results, it is shown that the RSCU values of the preference codons are all greater than 1 (Figure 3). Furthermore, the RSCU analysis reveals that the codons of all the PCGs have a strong preference, as RSCU values of NNU and NNA (which correspond to the codon that have U or A on the third position) are mostly higher than 1, and the frequency of usage is higher.

### 3.3. Phylogenetic Analysis

The phylogenetic relationships of *P. leptodactylus* sample from Greece (OQ131122) in comparison with other species (five haplotypes retrieved from GenBank with accession numbers: NC_033509.1, NC_033504.1, NC_026560.1, KX279349.1, and KX279347.1 belonging to the same family (Astacidae) are depicted in the maximum likelihood dendrogram of Figure 4. More specifically, NC_033509.1, NC_033504.1, NC_026560.1, KX279349.1, and KX279347.1 correspond to *P. leniusculus*, *A. pallipes*, *A. torrentium*, *A. leptodactylus*, and *A. astacus*, respectively. After alignment, these haplotypes were trimmed and eventually only sequences of the same length were compared. Further, branch lengths were included in the phylogenetic tree to evaluate the differences in the lengths of the branches.

## 4. Discussion

*P. leptodactylus* is a species with no clear systematic classification due to its various different morphological characteristics. Its most prevalent classification at the genus level is *P. leptodactylus* [19]. However, in Greece, there are limited data regarding the genetic identity of this species.

Even among the same basin, in the Caspian Sea the genetic structure of seven populations revealed significant haplotype diversity and nucleotide diversity [32]. Furthermore, many studies report the genetic variation existing among freshwater crayfish populations in many lakes and rivers [10,11,32,39,40,41]. This observation could be expected, as lakes and rivers operate as natural barriers for the population leading to distinct phylogenetic lineages, in contrast with marine species where typically low levels of genetic differentiation are observed mainly due to lack of natural boundaries and the high levels of dispersal potential [42].

The translocations that took place during the last years to new habitats were carried out by humans, where the narrow-clawed crayfish established populations. In most of these cases, there is a co-occurrence of *P. leptodactylus* with other freshwater crayfish species, such as *A. astacus* [10,43] and *A. torrentium* [5,9]. Therefore, the co-occurrence of *P. leptodactylus* with other freshwater species sharing common habitats can probably led to inbreeding and potential hybrids [11,23]. Therefore, it is of high importance to determine its genetic profile, as its populations are well-established in the new habitats after translocations. Hence, apart from the existing limited genetic data, after the translocation events, there is a major question rising regarding the assumption if this species developed distinct genetic identity in the new habitats. It is possible for *P. leptodactylus* to survive in the places where it has been transferred and established populations. From all the above, it is clear that these data from the mitochondrial genome sequence will provide useful information regarding translocations events for future biogeography analysis.

The present study, to the best of our knowledge, represents the first attempt to analyze the mitochondrial genome of *P. leptodactylus* in Greece, as well as representing the first step towards the investigation of its phylogenetic position based on the entire mitochondrial genome. The complete coding mitogenome of *P. leptodactylus* was submitted to GenBank, and given the Accession number OQ131122. Furthermore, we report 15 primer pairs that can be used for both *P. leptodactylus* and *A. astacus* for further analyses of different mitochondrial segments. The majority of *P. leptodactylus* genetic studies, so far, are mainly focused on genetic markers such as allozyme, single mitochondrial genes, and microsatellites [11,23,24,25] and limited information exists regarding the analysis of whole mitogenome. Furthermore, this specific genetic tool (15 primer sets and PCR conditions), as well as being cheaper, is also more user-friendly regarding the analysis process in comparison with Illumina method.

We observed that *P. leptodactylus* from Greece is genetically closer to *A. astacus*, and, indeed, has a smaller genetic distance compared to that of *A. torrentium* and *A. pallipes*, which belong to the same genus. In addition, the genetic distance between the analyzed sample and the unique sequence of *P. leptodactylus* submitted to GenBank (KX279349.1) is quite high, even though the individuals belong to the same species. The above observation suggests that the years after its translocation and population establishment, together with potential introgression with other species, has probably exerted an impact on the mitochondrial genome of wild *P. leptodactylus* individuals, leading to a distinct genetic identity after local adaptation. Of course, further research at population genetic level, including other Greek habitats where crayfish occur naturally or have been translocated, is necessary to draw conclusions regarding the genetic profile of the species and the potential existence of founder effects or cryptic species. Determination and enlightening of stock complexity at different locations is crucial for efficient species management strategies and stock assessment for aquaculture development [44,45]. Further, our results do not fully support the current taxonomic evaluation of *Astacus* and *Pontastacus* genera, which is based either on partial genes or on morphological data. Instead, complete mitochondrial-genome-based analysis seems to question the assignment of *Astacus* and *Pontastacus* in different genera, as well as raises concerns regarding the genetic distance of introduced populations in comparison with native ones.

Genetic analysis of the populations inhabiting Greece could lead to proper management in cases where the natural populations decline, omitting negative impacts in the ecosystem. Freshwater habitats, due to the existence of natural barriers, are characterized by restricted gene flow. Thus, the limited gene flow in combination with the co-occurrence of different freshwater crayfish species could result in different genetic structures as the years pass by. The improper management of translocations and restocking could lead to reduced genetic diversity and increased inbreeding levels. From all the above, it can be assumed that the genetic characterization of *P. leptodactylus* in Greece is of major importance for three main reasons: firstly, for a more efficient management of bloodstocks towards restocking purposes when populations decline; secondly, as a useful tool for aquaculture, as it can be used for genetic diversity, bloodstock management, and species identification [46]; and thirdly, for shedding light on the phylogenetic status of Greek populations inhabiting both natural and translocated habitats.

## 5. Conclusions

Here, we provide the first analysis of mitogenome of *P. leptodactylus* from Greece. It should be mentioned at this point that although one *P. leptodactylus* mitochondrial complete genome haplotype was already available in GenBank database, to the best of our knowledge, there was no study discussing this submission. Thus, this is the first study that describes the *P. leptodactylus* mitochondrial genome as well. The analyzed mitogenome of *P. leptodactylus* consists of 15,050 base pairs including 13 protein-coding genes, 2 rRNAs, and 22 tRNAs. From the phylogenetic relationships between the species studied, it is revealed that *P. leptodactylus* and *A. astacus* are closer than *A. torrentium* and *A. pallipes*, although the latter are classified in the same genera. Furthermore, high genetic distance is observed between the sample from Greece with the sequence existing in the GenBank. Hence, this partial mitogenome, together with the primer sets we describe, can operate as important and valuable information to support ongoing comparative genomic, phylogenomics, and molecular-based breeding studies for aquaculture, conservation, and biodiversity-related studies and can be improved upon over time, with the generation of additional data. To conclude, in addition to providing a fully annotated coding mitogenome of *P. leptodactylus* depicting its phylogenetic position, we developed a useful set of primers for further comprehensive analysis.

## Figures and Tables

**Figure 1 genes-14-00494-f001:**
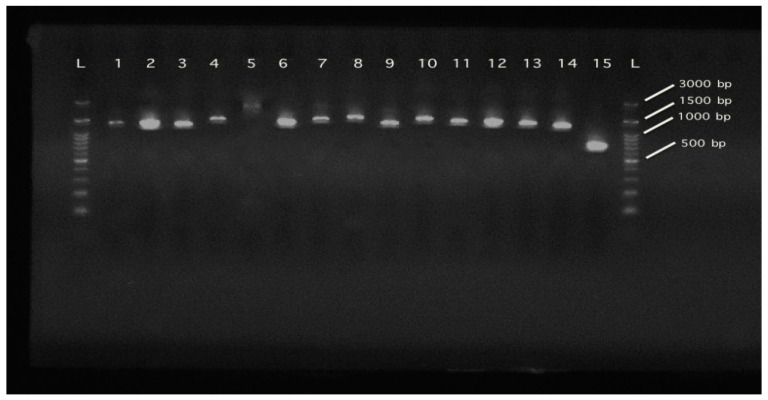
The fifteen PCR products from one individual amplified. Primer pair numbers according to Table 1. L: 100 bp Ladder (H3 RTU, GeneDireX).

**Figure 2 genes-14-00494-f002:**
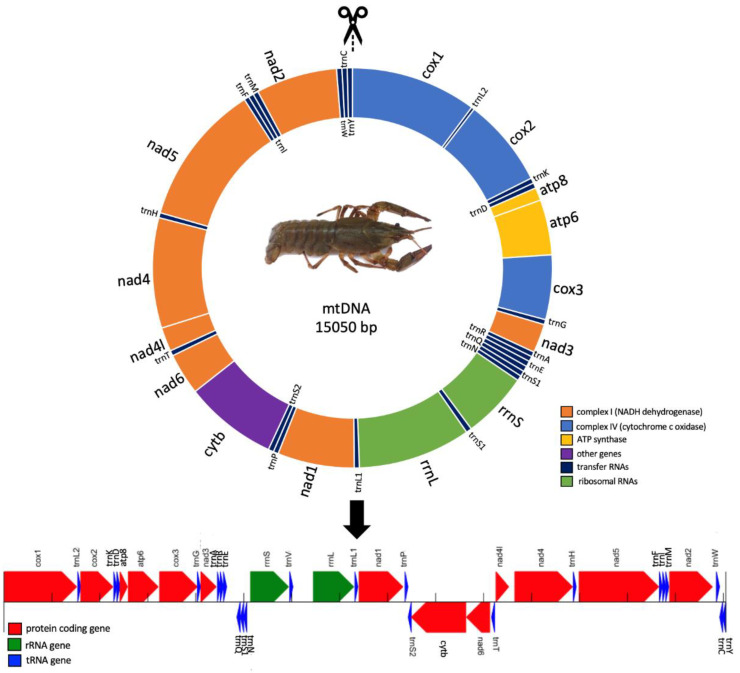
The annotated mitogenome of *Pontastacus leptodactylus*. Protein-coding, rRNA, and tRNA genes are shown with standard abbreviations and different colors. Arrows indicate transcription directions. The genes shown above the line indicate that the genes are on the direct strand, while genes depicted under the line are located on the reverse strand.

**Figure 3 genes-14-00494-f003:**
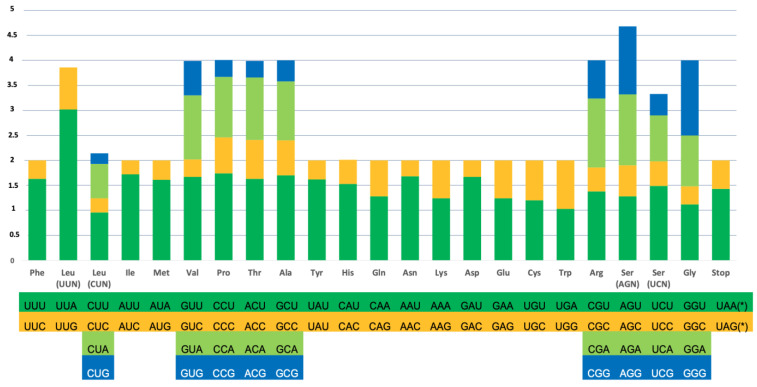
RSCU in the *Pontastacus leptodactylus* mitogenome. RSCU values are depicted on the *y*-axis while on the *x*-axis the codon families for amino acids and stop codons are shown. The asterisk “*” indicates the stop codon.

**Figure 4 genes-14-00494-f004:**
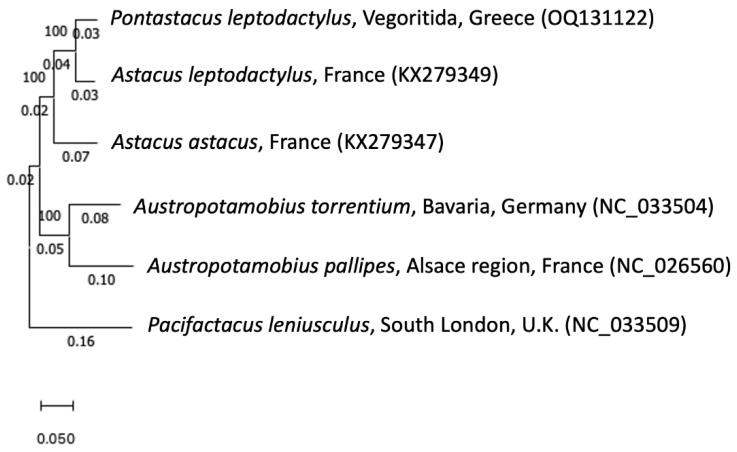
Maximum likelihood dendrogram depicting the phylogenetic position of Greek haplotype among the species *Pontastacus leptodactylus*, *Astacus astacus*, *Austropotamobius torrentium*, *Austropotamobius pallipes*, and *Pacifastacus leniusculus*.

**Table 1 genes-14-00494-t001:** Primer sets designed for the sequencing of mtDNA of *Pontastacus leptodactylus*.

Primer Set	Name	Sequence	Tm (°C)	Product Length (bp)	Genes Included
1	MtAst1F MtAst1R	5′ CAAATCATAAAGATATTGGAAC 3′ 5′ AATGTTGRGGRAAGAATGT 3′	49	1259	*coxI* partial
2	MtAst2F MtAst2R	5′ CAGTKGGRGGTTTAACAGGAG 3′ 5′ GTTATCRRGTGATTATTCTGAAC 3′	54	1255	*coxI* (partial) *trnL2* *coxII* *trnK* (partial)
3	MtAst3F MtAst3R	5′ CATTCTTGAACTGTACCTTC 3′ 5′ CTACTAAATGATATGCATGGTG 3′	52	1201	*coxII* (partial) *trnK* *trnD* *atp8* *atp6* *coxIII* (partial)
4	MtAst4.2F MtAst4.2R	5′ GCTGTTGCAATTATTCAGTC 3′ 5′ CCAAAGGTATAAGAAGMGTA 3′	50	1394	*atp6* (partial) *coxIII* *trnG* *nad3* (partial)
5	MtAst5F MtAst5R	5′ GGCTTCCAACCAAAAGGTC 3′ 5′ AGTWTAACCGCGACTGCTG 3′	49	~1500–2000	D-loop
6	MtAst6F MtAst6R	5′ GAATTTAACCGCTCAAGAAC 3′ 5′ CCTAACTATTTCTCTTCCGAG 3′	52	1287	*trnN* *rrnS* *trnV* *rrnL* (partial)
7	MtAst7F MtAst7R	5′ AGCATCTCATTTACACCGA 3′ 5′ ABTCBAACATGTCTAAGCATC 3′	51	1424	*rrnL* *trnL1* *nad1* (partial)
8	MtAst8F MtAst8R	5′ TACTTTAGGGATAACAGCGTA 3′ 5′ GGTCAAATTCTTTCACTCCT 3′	52	1502	*rrnL* (partial) *trnL1* *nad1* *trnP* *trnS2* *cytb* (partial)
9	MtAst9F MtAst9R	5′ TACCTCGGTTTCGTTATGA 3′ 5′ ATACYCCYAATATTGAATCAG 3′	51	1243	*nad1* (partial) *trnP* *trnS2* *cytb* (partial)
10	MtAst10F MtAst10R	5′ GACCTCARGGTAAGACATATC 3′ 5′ TCTCTCCCTAAYTGATTTCC 3′	54	1463	*cytb* (partial) *nad6* *trnT* *nad4l*
11	MtAst11F MtAst11R	5′ GTCCGCTCRCARGGTAATG 3′ 5′ AAAGGAAGYCAATGAAGAC 3′	51	1339	*nad4l* (partial) *nad4* *trnH* (partial)
12	MtAst12F MtAst12R	5′ GTCATGGTTTATGTTCATCTG 3′ 5′ ACTCAAAATTAGCTCCRAGC 3′	52	1285	*nad4l* (partial) *nad4* *trnH* *nad5* (partial)
13	MtAst13F MtAst13R	5′ CGTGTCRGCATTAGTACATTC 3′ 5′ CCCCAAAYCAAGAATTTGAAG 3′	54	1377	*nad5* (partial) *trnF* *trnI* *trnM* *nad2* (partial)
14	MtAst14F MtAst14R	5′ CTACATTGAAGCTGTAGAAGAG 3′ 5′ TTTGACAACTTTGAAGGATG 3′	51	1213	*trnW* (partial) *nad2* *trnW* (partial)
15	MtAst15F MtAst15R	5′ TCAGCAGGCCTATCATTT 3′ 5′ CAAAAGCATGAGCAGTTACTAC 3′	51	668	*nad2* (partial) *trnW* *trnC* *trnY* *coxI* (partial)

**Table 2 genes-14-00494-t002:** Annotation of the coding mitogenome of *Pontastacus leptodactylus*.

Position	Gene	Name	Strand	Start Codon	Stop Codon	Anticodon	Gene Length/bp
1–1537	Cytochrome c oxidase subunit 1	*cox1*	+	ATT	TTA		1537
1539–1601	tRNA	*trnL2*	+			TAA	63
1602–2288	Cytochrome c oxidase subunit 2	*cox2*	+	ATG	TAA		687
2290–2353	tRNA	*trnK*	+			TTT	64
2355–2420	tRNA	*trnD*	+			GTC	66
2421–2579	ATP synthase F0 subunit 8	*atp8*	+	GTG	TAG		159
2573–3247	ATP synthase F0 subunit 6	*atp6*	+	ATG	TAA		675
3247–4035	Cytochrome c oxidase subunit 3	*cox3*	+	ATG	TAA		789
4034–4095	tRNA	*trnG*	+			TCC	62
4096–4449	NADH dehydrogenase subunit 3	*nad3*	+	ATT	TAA		354
4451–4512	tRNA	*trnA*	+			TGC	62
4511–4575	tRNA	*trnR*	+			TCG	65
4576-4645	tRNA	*trnE*	+			TTC	70
~1500–2000 base pairs	Gap	*dloop*					
4858–4926	tRNA	*trnQ*	−			TTG	69
4935–5001	tRNA	*trnS1*	−			TCT	67
5002–5065	tRNA	*trnN*	−			GTT	64
5143–5956	rRNA	*rrnS*	+				814
5959–6027	tRNA	*trnV*	+			TAC	69
5993–7371	rRNA (partial)	*rrnL*	+				1379
7318–7380	tRNA	*trnL1*	+			TAG	63
7405–8343	NADH dehydrogenase subunit 1	*nad1*	+	ATA	TAA		939
8363–8424	tRNA	*trnP*	+			TGG	62
8429–8495	tRNA	*trnS2*	−			TGA	67
8496–9630	Cytochrome b	*cytb*	−	ATG	ATT		1135
9630–10,139	NADH dehydrogenase subunit 6	*nad6*	−	ATT	TAA		510
10,167–10,232	tRNA	*trnT*	−			TGT	66
10,235–10,528	NADH dehydrogenase subunit 4	*nad4L*	+	ATG	TAA		294
10,525–11,871	NADH dehydrogenase subunit 4	*nad4*	+	ATA	TAA		1347
11,871–11,932	tRNA	*trnH*	+			GTG	62
11,933–13,666	NADH dehydrogenase subunit 5	*nad5*	+	ATG	TAA		1734
13,666–13,727	tRNA	*trnF*	+			GAA	62
13,735–13,798	tRNA	*trnI*	+			GAT	64
13,798–13,862	tRNA	*trnM*	+			CAT	65
13,863–14,855	NADH dehydrogenase subunit 2	*nad2*	+	GTG	TAA		993
14,855–14,921	tRNA	*trnW*	+			TCA	67
14,923–14,986	tRNA	*trnC*	−			GCA	64
14,987–15,050	tRNA	*trnY*	−			GTA	64

## Data Availability

The data presented in this study are openly available in GenBank^®^ NIH genetic sequence database under the accession number OQ131122.

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
