# Peer review of "The Coding Mitogenome of the Freshwater Crayfish Pontastacus leptodactylus (Decapoda:Astacidea:Astacidae) from Lake Vegoritida, Greece and Its Taxonomic Classification"

_genes, 2023, doi:10.3390/genes14020494_

Round 1

Reviewer 1 Report

The paper presents the first mitogenome of Pontastacus leptodactylus and shows some comparative analysis to close decapod crayfish species.

I ask the authors to define the number of crayfish used for this work, because it appears from the M&M that only one was taken. So, this part should be better clarified.

If 14 primer sets were used, why do the authors mention 15 PCR products in line 92?

The authors should run a gel with all the PCR products together and not arranged in different images, also appropriate ladders should be used.

The phylogenetic analysis to evaluate the distances between the compared mitogenomes should include a branch length test to evaluate the statistical difference in the length of the branches. In addition, and more importantly partial and complete mitochondrial sequences are compared in Figure 3 and it is not clear if sequences of the same length were compared. If the sequences are of different lengths the analysis should be performed using the same length, and if only overlapping regions were analysed, authors should indicate which portion of the mitogenome was used for phylogenetic analysis.

At the moment, the paper has many flaws and parts of it should be clarified or rewritten. The current state of the paper requires a major revision.

Minor revisions:

Line 85: authors should choose to mention the number of primer sets used in letters or in numbers.

Line 93: when mentioning the final pmol used for each primer, authors should relate pmol to a volume. In this part, however, authors use: an initial volume to set the amount of ready mix used, then a final concentration of primers and then 50 ng of what, I guess DNA, but this information is lacking. The whole part should be rewritten.

Line 95: “hybridization temperature” should be replaced by “annealing” or “melting” temperature.

Lines 98-99: if authors mention bidirectional sequencing, they do not need to write in the next line that used bot forward and reverse primers to get Sanger sequencing.

Figure 1a: the electrophoretic gel is overloaded both for the PCR products that for the ladder. This figure is fine for a lab notebook, but not for a publication. Authors should re-run a gel loading less amount of samples into the wells.

Lines 110-114: Moreover, authors mentioned 15 mt PCR products, 14 of them successfully sequenced, but only 13 samples are present in the gel figure. Numbers do not coincide.

Figure 1b: authors should use 2 or 3 Kb ladders, this figure with amplicons greater than the ladder is unacceptable.

Figure 3: species names in the figure should be italicised or authors might use codes. Moreover, if the first sequence reports the place, also for other sequences geographic place should be included rather then the partial or complete fragment the sequences derived from.

Author Response

Comments and Suggestions for Authors: The paper presents the first mitogenome of Pontastacus leptodactylus and shows some comparative analysis to close decapod crayfish species.

I ask the authors to define the number of crayfish used for this work, because it appears from the M&M that only one was taken. So, this part should be better clarified.

Response: Five individuals were analyzed, but only one (the best annotated) was eventually included in this study. Our main goal apart from providing the phylogenetic position of this species is also to construct a tool (with the primer set and PCR conditions) for more comprehensive future phylogenetic investigations that is at the same time cheaper and easier to perform without the need of sophisticated bioinformatic platforms. This clarification was added in the Materials and Methods section (please see lines 218-219 and 234-235 in the revised manuscript).

If 14 primer sets were used, why do the authors mention 15 PCR products in line 92?

Response: We apologize for this mistake. Indeed 14 primer sets was not the correct number, which was corrected by “15” as also indicated in Table 1 and Figure 1

(please see line 224 in the revised manuscript).

The authors should run a gel with all the PCR products together and not arranged in different images, also appropriate ladders should be used.

Response: In accordance to the reviewer’s comment, a new run was conducted in an agarose gel with all the PCR products together and an appropriate new ladder, as depicted in Figure 1 in the revised manuscript. Hence, Figures 1a,b,c were deleted (please see lines 333-334 for figure 1 in the revised manuscript).

The phylogenetic analysis to evaluate the distances between the compared mitogenomes should include a branch length test to evaluate the statistical difference in the length of the branches. In addition, and more importantly partial and complete mitochondrial sequences are compared in Figure 3 and it is not clear if sequences of the same length were compared. If the sequences are of different lengths the analysis should be performed using the same length, and if only overlapping regions were analysed, authors should indicate which portion of the mitogenome was used for phylogenetic analysis.

Response: Branch lengths were included in the phylogenetic tree in order to evaluate the differences in the lengths of the branches as recommended by the reviewer. It should be also clarified that all sequences compared were trimmed after alignment and thus only sequences of the same length were compared (this was added for clarification in section 3.3 of the revised manuscript). The overlapping regions not included were correspondingly the D-loop region (please see lines 245-246 and 419-422 in the revised manuscript).

At the moment, the paper has many flaws and parts of it should be clarified or rewritten. The current state of the paper requires a major revision.

Response: We provided an in depth revision flaws were corrected according to both reviewers’ comments, whereas several parts were rewritten. Also a new analysis was added regarding the relative synonymous codon usage (please see Figure 3 in the revised manuscript).

Minor revisions:

Line 85: authors should choose to mention the number of primer sets used in letters or in numbers.

Response: As it is in the beginning of the sentence, we preferred to use only the letters (please see line 224 in the revised manuscript).

Line 93: when mentioning the final pmol used for each primer, authors should relate pmol to a volume. In this part, however, authors use: an initial volume to set the amount of ready mix used, then a final concentration of primers and then 50 ng of what, I guess DNA, but this information is lacking. The whole part should be rewritten.

Response: the whole part was rewritten, referring to the volume first. More specifically, the total reaction volume was 20ul. 10ul of Fast gene polymerase Ready mix 2X was used; 0,6 μl of each primer of 10μΜ;, 1μl of DNA sample (50ng/ul); and the rest of the volume (7,8 μl) was completed with ultrapure water (please see lines 230-232 in the revised manuscript).

Line 95: “hybridization temperature” should be replaced by “annealing” or “melting” temperature.

Response: “hybridization temperature” was replaced by “annealing temperature” as recommended by the reviewer (please see line 233 in the revised manuscript).

Lines 98-99: if authors mention bidirectional sequencing, they do not need to write in the next line that used bot forward and reverse primers to get Sanger sequencing.

Response: In accordance to the reviewer’s comment, “using both forward and reverse primers” was deleted (please see line 238 in the revised manuscript).

Figure 1a: the electrophoretic gel is overloaded both for the PCR products that for the ladder. This figure is fine for a lab notebook, but not for a publication. Authors should re-run a gel loading less amount of samples into the wells.

Response: Following the reviewer’s comment, we re-run a gel using less amount of samples and ladder into the wells, using another ladder and with all 15 products in the same gel, which is depicted in Figure 1 in the revised manuscript. Hence, figures 1a,b,c were deleted (please see Figure 1 in the revised manuscript).

Lines 110-114: Moreover, authors mentioned 15 mt PCR products, 14 of them successfully sequenced, but only 13 samples are present in the gel figure. Numbers do not coincide.

Response: Following the reviewer’s comment, we re-run a gel with all of 15 PCR products being present in the gel figure, which are depicted in Figure 1. The PCR products obtained after the reaction were 15, but only 14 of them were successfully sequenced. The product which failed to be sequenced included the D-loop region and was approximately 2000 bp in length as depicted in Figure 1 (please see figure 1 in the revised manuscript).

Figure 1b: authors should use 2 or 3 Kb ladders, this figure with amplicons greater than the ladder is unacceptable.

Response: Following the reviewer’s comment, we re-run a gel using a 3Kb ladder (100bp ladder with the bigger band reaching 3Kb). The figure is depicted in Figure 1, whereas Figure 1b was deleted (please see Figure 1 in the revised manuscript).

Figure 3: species names in the figure should be italicised or authors might use codes. Moreover, if the first sequence reports the place, also for other sequences geographic place should be included rather then the partial or complete fragment the sequences derived from.

Response: the species names in the figure were italicized and the accession number of each one has been also included. In addition, the geographic place for the other sequences (apart from the first which derived from the present study) were included. The fragment that the sequences derived from is no longer depicted on the branches of the dendrogram (please see Figure 4 in the revised manuscript).

Reviewer 2 Report

The aim of the present study was firstly to analyze the entire mitochondrial genome of the Pontastacus leptodactylus collected from Greece for the first time and further to attempt to clarify its systematic classification. The justification for the study is satisfactory.

 Based on the results obtained, the authors raise interesting systematic questions to be investigated in the future. However, some questions were unclear and could be better explained or revised, for example:

Why choose specimens of Lake Vegoritida? It would not be better to have worked with animal tissue from the type locality, in order to solve systematic issues?

 What is the distribution of Pontastacus leptodactylus, is it endemic to Greece, or does it occur in other countries?

What is the extent of occurrence of the species, or rather, is it present in independent watersheds? Information on the biogeography of the species needs to be included. Based on these data and sequencing the mitochondrial genome of the species, a phylogeographic analysis would be interesting.  

 Results

Figures 1a, 1b and 1c are unnecessary.

What percentage of the mitochondrial genome of Astacus leptodactylus and Astacus astacus species is missing? Can't this lack interfere with the phylogenetic results (Fig. 3)?

 Discussion

The text below is a repeat of what was written in introduction, it is unnecessary: “In Greece, P. leptodactylus is native only to the Evros region. However, during the last years many translocations of P. leptodactylus took place”.

 The sentence below is confusing:

“Therefore, combining the studies highlighting the high levels of genetic differentiation of P. leptodactylus with its co-occurrence with other freshwater species which probably led to in-breeding and potential hybrids, it is of high importance to determine its genetic profile, as its populations are well established in the new habitats after translocations, for the last few years.”  Did the authors mean genetic diversity in place of genetic differentiation? Because if there is high genetic differentiation, you wouldn't expect hybrids.

 The text below is a repeat of what was written in the introduction, it is unnecessary:

  “The complete coding mitogenome of P. leptodactylus consists of 15.050 base”.

 “Complete mitochondrial genomes are generally powerful tools that have been extensively used to study the phylogenetic relationships and evolution of metazoan organisms”.

Author Response

The aim of the present study was firstly to analyze the entire mitochondrial genome of the Pontastacus leptodactylus collected from Greece for the first time and further to attempt to clarify its systematic classification. The justification for the study is satisfactory.

Response: We are grateful to the 2nd reviewer for the constructive comments and the recognition of the importance of our manuscript. We hope that the level and quality of the revised version of our manuscript, following her/his suggestions, will satisfy the reviewer’s expectations.

Based on the results obtained, the authors raise interesting systematic questions to be investigated in the future. However, some questions were unclear and could be better explained or revised, for example:

Why choose specimens of Lake Vegoritida? It would not be better to have worked with animal tissue from the type locality, in order to solve systematic issues?

Response: The study of animal tissue from the type locality is in our future goals (and the subsequent comparison with the specimen from lake Vegoritida from the present study), in an upcoming project. This could provide information regarding the origin of the translocated individuals. We chose this lake (as the first step towards systematics clarification) because the species has been translocated there many years ago and there are questions regarding formation of distinct genetic identity as well as generation of hybrids since it co-occurrs with other species such as A. astacus and A. torrentium. This explanation was added in the scope paragraph of the Introduction section in the revised manuscript (please see lines 210-216 in the revised manuscript).

What is the distribution of Pontastacus leptodactylus, is it endemic to Greece, or does it occur in other countries?

Response: Pontastacus leptodactylus occurs in several counties (32), and is characterized by wide distribution. It is indigenous but not endemic in Greece. More specifically, it is indigenous to Europe and Anatolia. All the above information regarding its distribution was added in the introduction according to the reviewer’s comment (please see lines 63-66 and 85-90 in the revised manuscript).

What is the extent of occurrence of the species, or rather, is it present in independent watersheds? Information on the biogeography of the species needs to be included. Based on these data and sequencing the mitochondrial genome of the species, a phylogeographic analysis would be interesting.

Response: As this species is indigenous to both Europe and Anatolia, characterized by a wide distribution it is expected to be present in independent watersheds (i.e. in Evros and in Turkey). However, as it is the most economic important ICS species its distribution is highly affected by human translocations outside of its natural range. Furthermore, it is less sensitive to pollution, to salinity fluctuations and to diseases (such as crayfish plague that devastated A. astacus populations) and it can tolerate a broad range of temperatures. Hence, it is possible to survive in the places where it has been transferred and establish populations. From all the above, its clear that these data from its mitochondrial genome sequence will provide useful information regarding translocations events as well as for future biogeography analysis. This info has been added in both introduction and discussion sections (please see lines 41-47; 50-54; 63-66 and 85-90 in the revised manuscript).

Results

Figures 1a, 1b and 1c are unnecessary.

What percentage of the mitochondrial genome of Astacus leptodactylus and Astacus astacus species is missing? Can't this lack interfere with the phylogenetic results (Fig. 3)?

Response: Figures 1a, 1b, 1c were replaced with one figure, depicting all the PCR products together in one electrophoresis gel, with a ladder of bigger size (3Kb), indicating that the part missing is approximately 1,5-2Kb. In the phylogenetic results only sequences of the same size with aligned regions were included in the analysis. Due to the addition of a new figure, the phylogenetic results are depicted in Figure 4 (please see Figure 1 and Figure 4 on the revised manuscript).

Discussion

The text below is a repeat of what was written in introduction, it is unnecessary: “In Greece, P. leptodactylus is native only to the Evros region. However, during the last years many translocations of P. leptodactylus took place”.

Response: the repetition with native range of P. leptodactylus was deleted, and the sentence rephrased to: “The translocations that took place during the last years to new habitats, were carried out by humans, where the narrow-clawed crayfish established populations.” (please see lines 485-487 in the revised manuscript).

The sentence below is confusing:

“Therefore, combining the studies highlighting the high levels of genetic differentiation of P. leptodactylus with its co-occurrence with other freshwater species which probably led to in-breeding and potential hybrids, it is of high importance to determine its genetic profile, as its populations are well established in the new habitats after translocations, for the last few years.”

Did the authors mean genetic diversity in place of genetic differentiation? Because if there is high genetic differentiation, you wouldn't expect hybrids.

Response: Indeed, genetic diversity is more appropriate term at this point. The sentence was rewritten avoiding the confusing terms (please see lines 486-488 in the revised manuscript).

The text below is a repeat of what was written in the introduction, it is unnecessary: “The complete coding mitogenome of P. leptodactylus consists of 15.050 base”.

“Complete mitochondrial genomes are generally powerful tools that have been extensively used to study the phylogenetic relationships and evolution of metazoan organisms”.

Response: As these two sentences were unnecessary due to repetitions in the introduction they were deleted from the discussion in accordance to the reviewer’s comment.